# Fast Approximate Natural Gradient Descent in a Kronecker-factored Eigenbasis

**Thomas George**[*1]**, César Laurent**[*1]**, Xavier Bouthillier**[1]**, Nicolas Ballas**[2]**, Pascal Vincent**[1,2,3]

[1] Mila - Université de Montréal;    [2] Facebook AI Research;    [3] CIFAR;    * *equal contribution*
{thomas.george, cesar.laurent, xavier.bouthillier}@umontreal.ca
{ballasn, pascal}@fb.com

## Abstract

Optimization algorithms that leverage gradient covariance information, such as variants of natural gradient descent (Amari, 1998), offer the prospect of yielding more effective descent directions. For models with many parameters, the covariance matrix they are based on becomes gigantic, making them inapplicable in their original form. This has motivated research into both simple diagonal approximations and more sophisticated factored approximations such as KFAC (Heskes, 2000; Martens & Grosse, 2015; Grosse & Martens, 2016). In the present work we draw inspiration from both to propose a novel approximation that is provably better than KFAC and amendable to cheap partial updates. It consists in tracking a diagonal variance, not in parameter coordinates, but in a Kronecker-factored eigenbasis, in which the diagonal approximation is likely to be more effective. Experiments show improvements over KFAC in optimization speed for several deep network architectures.

## 1    Introduction

Deep networks have exhibited state-of-the-art performance in many application areas, including image recognition (He et al., 2016) and natural language processing (Gehring et al., 2017). However top-performing systems often require days of training time and a large amount of computational power, so there is a need for efficient training methods.

Stochastic Gradient Descent (SGD) and its variants are the current workhorse for training neural networks. Training consists in optimizing the network parameters $\theta$ (of size $n_\theta$) to minimize a regularized empirical risk $R(\theta)$, through gradient descent. The negative loss gradient is approximated based on a small subset of training examples (a mini-batch). The loss functions of neural networks are highly non-convex functions of the parameters, and the loss surface is known to have highly imbalanced curvature which limits the efficiency of $1^{st}$ order optimization methods such as SGD.

Methods that employ $2^{nd}$ order information have the potential to speed up $1^{st}$ order gradient descent by correcting for imbalanced curvature. The parameters are then updated as: $\theta \leftarrow \theta - \eta G^{-1} \nabla_\theta R(\theta)$, where $\eta$ is a positive learning-rate and $G$ is a preconditioning matrix capturing the local curvature or related information such as the Hessian matrix in Newton's method or the Fisher Information Matrix in Natural Gradient (Amari, 1998). Matrix $G$ has a gigantic size $n_\theta \times n_\theta$ which makes it too large to compute and invert in the context of modern deep neural networks with millions of parameters. For practical applications, it is necessary to trade-off quality of curvature information for efficiency.

A long family of algorithms used for optimizing neural networks can be viewed as approximating the diagonal of a large preconditioning matrix. Diagonal approximations of the Hessian (Becker et al., 1988) have been proven to be efficient, and algorithms that use the diagonal of the covariance matrix of the gradients are widely used among neural networks practitioners, such as Adagrad (Duchi et al.,

2011), Adadelta (Zeiler, 2012), RMSProp (Tieleman & Hinton, 2012), Adam (Kingma & Ba, 2015). We refer the reader to Bottou et al. (2016) for an informative review of optimization methods for deep networks, including diagonal rescalings, and connections with the Batch Normalization (BN) (Ioffe & Szegedy, 2015) technique.

More elaborate algorithms do not restrict to diagonal approximations, but instead aim at accounting for some correlations between different parameters (as encoded by non-diagonal elements of the preconditioning matrix). These methods range from Ollivier (2015) who introduces a rank 1 update that accounts for the cross correlations between the biases and the weight matrices, to quasi Newton methods (Liu & Nocedal, 1989) that build a running estimate of the exact non-diagonal preconditioning matrix, and also include block diagonals approaches with blocks corresponding to entire layers (Heskes, 2000; Desjardins et al., 2015; Martens & Grosse, 2015; Fujimoto & Ohira, 2018). Factored approximations such as KFAC (Martens & Grosse, 2015; Ba et al., 2017) approximate each block as a Kronecker product of two much smaller matrices, both of which can be estimated and inverted more efficiently than the full block matrix, since the inverse of a Kronecker product of two matrices is the Kronecker product of their inverses.

In the present work, we draw inspiration from both diagonal and factored approximations. We introduce an Eigenvalue-corrected Kronecker Factorization (EKFAC) that consists in tracking a diagonal variance, not in parameter coordinates, but in a Kronecker-factored eigenbasis. We show that EKFAC is a provably better approximation of the Fisher Information Matrix than KFAC. In addition, while computing the Kronecker-factored eigenbasis is a computationally expensive operation that needs to be amortized, tracking of the diagonal variance is a cheap operation. EKFAC therefore allows to perform partial updates of our curvature estimate $G$ at the iteration level. We conduct an empirical evaluation of EKFAC on the deep auto-encoder optimization task using fully-connected networks and CIFAR-10 relying on deep convolutional neural networks where EKFAC shows improvements over KFAC in optimization.

## 2    Background and notations

We are given a dataset $\mathcal{D}_{\mathrm{train}}$ containing (input, target) examples $(x, y)$, and a neural network $f_\theta(x)$ with parameter vector $\theta$ of size $n_\theta$. We want to find a value of $\theta$ that minimizes an empirical risk $R(\theta)$ expressed as an average of a loss $\ell$ incurred by $f_\theta$ over the training set: $R(\theta) = \mathbb{E}_{(x,y) \in \mathcal{D}_{\mathrm{train}}} [\ell(f_\theta(x), y)]$. We will use $\mathbb{E}$ to denote both expectations w.r.t. a distribution or, as here, averages over finite sets, as made clear by the subscript and context. Considered algorithms for optimizing $R(\theta)$ use stochastic gradients $\nabla_\theta = \nabla_\theta(x, y) = \frac{\partial \ell(f_\theta(x), y)}{\partial \theta}$, or their average over a mini-batch of examples $\mathcal{D}_{\mathrm{mini}}$ sampled from $\mathcal{D}_{\mathrm{train}}$. Stochastic gradient descent (SGD) does a $1^{\mathrm{st}}$ order update: $\theta \leftarrow \theta - \eta \nabla_\theta$ where $\eta$ is a positive learning rate. $2^{\mathrm{nd}}$ order methods first multiply $\nabla_\theta$ by a preconditioning matrix $G^{-1}$ yielding the update: $\theta \leftarrow \theta - \eta G^{-1} \nabla_\theta$. Preconditioning matrices for Natural Gradient (Amari, 1998) / Generalized Gauss-Newton (Schraudolph, 2001) / TONGA (Le Roux et al., 2008) can all be expressed as either (centered) covariance or (uncentered) second moment of $\nabla_\theta$, computed over slightly different distributions of $(x, y)$. The natural gradient uses the Fisher Information Matrix, which for a probabilistic classifier can be expressed as $G = \mathbb{E}_{x \in \mathcal{D}_{\mathrm{train}}, y \sim p_\theta(\mathbf{y}|x)} \left[ \nabla_\theta \nabla_\theta^\top \right]$ where the expectation is taken over targets sampled form the *model* $p_\theta = f_\theta$. By contrast, the *empirical Fisher* approximation or generalized Gauss-Newton uses $G = \mathbb{E}_{(x,y) \in \mathcal{D}_{\mathrm{train}}} \left[ \nabla_\theta \nabla_\theta^\top \right]$. Our discussion and development applies regardless of the precise distribution over $(x, y)$ used to estimate a $G$ so we will from here on use $\mathbb{E}$ without a subscript.

Matrix $G$ has a gigantic size $n_\theta \times n_\theta$, which makes it too big to compute and invert. In order to get a practical algorithm, we must find approximations of $G$ that keep some of the relevant $2^{\mathrm{nd}}$ order information while removing the unnecessary and computationally costly parts. A first simplification, adopted by nearly all prior approaches, consists in treating each layer of the neural network separately, ignoring cross-layer terms. This amounts to a first block-diagonal approximation of $G$: each block $G^{(l)}$ caters for the parameters of a single layer $l$. Now $G^{(l)}$ can typically still be extremely large.

A cheap but very crude approximation consists in using a diagonal $G^{(l)}$, i.e. taking into account the variance in each parameter dimension, but ignoring all covariance structure. A less stringent approximation was proposed by Heskes (2000) and later Martens & Grosse (2015). They propose to approximate $G^{(l)}$ as a Kronecker product $G^{(l)} \approx A \otimes B$ which involves two smaller matrices,

making it much cheaper to store, compute and invert[1]. Specifically for a layer $l$ that receives input $h$ of size $d_{\text{in}}$ and computes linear pre-activations $a = W^\top h$ of size $d_{\text{out}}$ (biases omitted for simplicity) followed by some non-linear activation function, let the backpropagated gradient on $a$ be $\delta = \frac{\partial \ell}{\partial a}$. The gradients on parameters $\theta^{(l)} = W$ will be $\nabla_W = \frac{\partial \ell}{\partial W} = \text{vec}(h\delta^\top)$. The Kronecker factored approximation of corresponding $G^{(l)} = \mathbb{E}\left[\nabla_W \nabla_W^\top\right]$ will use $A = \mathbb{E}\left[hh^\top\right]$ and $B = \mathbb{E}\left[\delta\delta^\top\right]$ i.e. matrices of size $d_{\text{in}} \times d_{\text{in}}$ and $d_{\text{out}} \times d_{\text{out}}$, whereas the full $G^{(l)}$ would be of size $d_{\text{in}}d_{\text{out}} \times d_{\text{in}}d_{\text{out}}$. Using this Kronecker approximation (known as KFAC) corresponds to approximating entries of $G^{(l)}$ as follows: $G^{(l)}_{ij,i'j'} = \mathbb{E}\left[\nabla_{W_{ij}} \nabla_{W_{i'j'}}^\top\right] = \mathbb{E}\left[(h_i\delta_j)(h_{i'}\delta_{j'})\right] \approx \mathbb{E}\left[h_ih_{i'}\right]\mathbb{E}\left[\delta_j\delta_{j'}\right]$.

A similar principle can be applied to obtain a Kronecker-factored expression for the covariance of the gradients of the parameters of a convolutional layer (Grosse & Martens, 2016). To obtain matrices $A$ and $B$ one then needs to also sum over spatial locations and corresponding receptive fields, as illustrated in Figure 1.

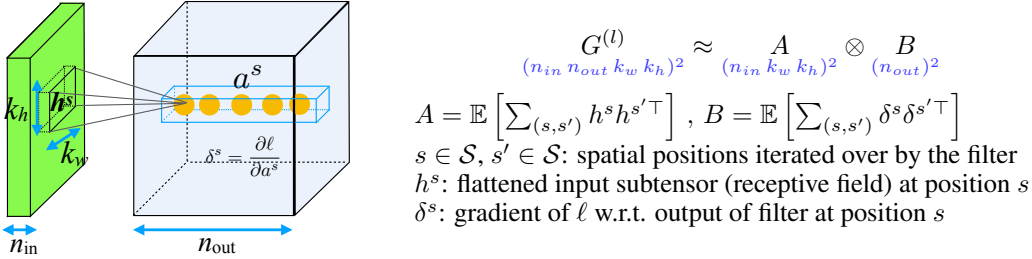

$$G^{(l)}_{(n_{in}\,n_{out}\,k_w\,k_h)^2} \approx A_{(n_{in}\,k_w\,k_h)^2} \otimes B_{(n_{out})^2}$$

$A = \mathbb{E}\left[\sum_{(s,s')} h^s h^{s'\top}\right]$ , $B = \mathbb{E}\left[\sum_{(s,s')} \delta^s \delta^{s'\top}\right]$

$s \in \mathcal{S}$, $s' \in \mathcal{S}$: spatial positions iterated over by the filter
$h^s$: flattened input subtensor (receptive field) at position $s$
$\delta^s$: gradient of $\ell$ w.r.t. output of filter at position $s$

Figure 1: KFAC for convolutional layer with $n_{\text{out}}n_{\text{in}}k_wk_h$ parameters.

# 3  Proposed method

## 3.1  Motivation: reflexion on diagonal rescaling in different coordinate bases

It is instructive to contrast the type of "exact" natural gradient preconditioning of the gradient that uses the full Fisher Information Matrix would yield, to what we do when approximating this by using a diagonal matrix only. Using the full matrix $G = \mathbb{E}[\nabla_\theta \nabla_\theta^\top]$ yields the natural gradient update: $\theta \leftarrow \theta - \eta G^{-1}\nabla_\theta$. When resorting to a diagonal approximation we instead use $G_{\text{diag}} = \text{diag}(\sigma_1^2, \ldots, \sigma_{n_\theta}^2)$ where $\sigma_i^2 = G_{i,i} = \mathbb{E}[(\nabla_\theta)_i^2]$. So that update $\theta \leftarrow \theta - \eta G_{\text{diag}}^{-1}\nabla_\theta$ amounts to preconditioning the gradient vector $\nabla_\theta$ by dividing each of its coordinates by an estimated second moment $\sigma_i^2$. This diagonal rescaling happens in the initial basis of parameters $\theta$. By contrast, a full natural gradient update can be seen to do a similar diagonal rescaling, not along the initial parameter basis axes, but along the axes of the *eigenbasis* of the matrix $G$. Let $G = USU^\top$ be the eigendecomposition of $G$. The operations that yield the full natural gradient update $G^{-1}\nabla_\theta = US^{-1}U^\top\nabla_\theta$ correspond to the sequence of a) multiplying gradient vector $\nabla_\theta$ by $U^\top$ which corresponds to switching to the eigenbasis: $U^\top\nabla_\theta$ yields the coordinates of the gradient vector expressed in that basis b) multiplying by a diagonal matrix $S^{-1}$, which rescales each coordinate $i$ (in that eigenbasis) by $S_{ii}^{-1}$ c) multiplying by $U$, which switches the rescaled vector back to the initial basis of parameters. It is easy to show that $S_{ii} = \mathbb{E}[(U^\top\nabla_\theta)_i^2]$ (proof is given in Appendix A.2). So similarly to what we do when using a diagonal approximation, we are rescaling by the second moment of gradient vector components, but rather than doing this in the initial parameter basis, we do this in the eigenbasis of $G$. Note that the variance measured along the leading eigenvector can be much larger than the variance along the axes of the initial parameter basis, so the effects of the rescaling by using either the full $G$ or its diagonal approximation can be very different.

Now what happens when we use the less crude KFAC approximation instead? We approximate[2] $G \approx A \otimes B$ yielding the update $\theta \leftarrow \theta - \eta(A \otimes B)^{-1}\nabla_\theta$. Let us similarly look at it through its eigendecomposition. The eigendecomposition of the Kronecker product $A \otimes B$ of two real symmetric

positive semi-definite matrices can be expressed using their own eigendecomposition $A = U_A S_A U_A^\top$ and $B = U_B S_B U_B^\top$, yielding $A \otimes B = (U_A S_A U_A^\top) \otimes (U_B S_B U_B^\top) = (U_A \otimes U_B)(S_A \otimes S_B)(U_A \otimes U_B)^\top$. $U_A \otimes U_B$ gives the orthogonal eigenbasis of the Kronecker product, we call it the Kronecker-Factored Eigenbasis (KFE). $S_A \otimes S_B$ is the diagonal matrix containing the associated eigenvalues. Note that each such eigenvalue will be a product of an eigenvalue of $A$ stored in $S_A$ and an eigenvalue of $B$ stored in $S_B$. We can view the action of the resulting Kronecker-factored preconditioning in the same way as we viewed the preconditioning by the full matrix: it consists in a) expressing gradient vector $\nabla_\theta$ in a different basis $U_A \otimes U_B$ which can be thought of as approximating the directions of $U$, b) doing a diagonal rescaling by $S_A \otimes S_B$ *in that basis,* c) switching back to the initial parameter space. Here however the rescaling factor $(S_A \otimes S_B)_{ii}$ is *not* guaranteed to match the second moment along the associated eigenvector $\mathbb{E}[((U_A \otimes U_B)^\top \nabla_\theta)_i^2]$.

In summary (see Figure 2):

- Full matrix $G$ preconditioning will scale by variances estimated along the eigenbasis of $G$.
- Diagonal preconditioning will scale by variances properly estimated, but along the initial parameter basis, which can be very far from the eigenbasis of $G$.
- KFAC preconditioning will scale the gradient along the KFE basis that will likely be closer to the eigenbasis of $G$, but doesn't use properly estimated variances along these axes for this scaling (the scales being themselves constrained to being a Kronecker product $S_A \otimes S_B$).

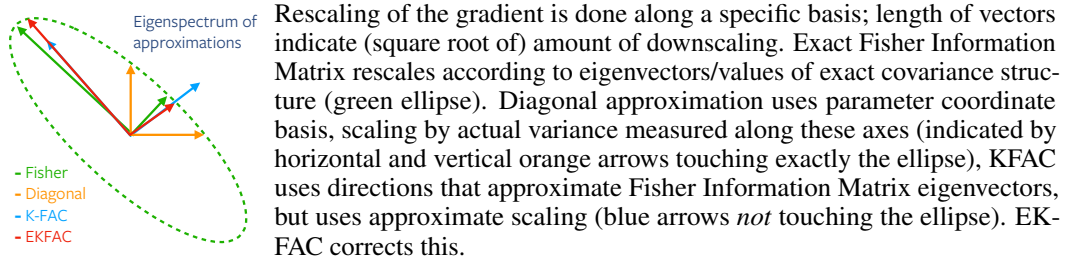

Eigenspectrum of approximations

- Fisher
- Diagonal
- K-FAC
- EKFAC

Rescaling of the gradient is done along a specific basis; length of vectors indicate (square root of) amount of downscaling. Exact Fisher Information Matrix rescales according to eigenvectors/values of exact covariance structure (green ellipse). Diagonal approximation uses parameter coordinate basis, scaling by actual variance measured along these axes (indicated by horizontal and vertical orange arrows touching exactly the ellipse), KFAC uses directions that approximate Fisher Information Matrix eigenvectors, but uses approximate scaling (blue arrows *not* touching the ellipse). EKFAC corrects this.

Figure 2: Cartoon illustration of rescaling achieved by different preconditioning strategies

## 3.2 Eigenvalue-corrected Kronecker Factorization (EKFAC)

To correct for the potentially inexact rescaling of KFAC, and obtain a better but still computationally efficient approximation, instead of $G_{\text{KFAC}} = A \otimes B = (U_A \otimes U_B)(S_A \otimes S_B)(U_A \otimes U_B)^\top$ we propose to use an *Eigenvalue-corrected Kronecker Factorization*: $G_{\text{EKFAC}} = (U_A \otimes U_B)S^*(U_A \otimes U_B)^\top$ where $S^*$ is the diagonal matrix defined by $S_{ii}^* = s_i^* = \mathbb{E}[((U_A \otimes U_B)^\top \nabla_\theta)_i^2]$. Vector $s^*$ is the vector of second moments of the gradient vector coordinates expressed in the Kronecker-factored Eigenbasis (KFE) $U_A \otimes U_B$ and can be efficiently estimated and stored.

In Appendix A.1 we prove that this $S^*$ is the optimal diagonal rescaling in that basis, in the sense that $S^* = \arg\min_S \|G - (U_A \otimes U_B)S(U_A \otimes U_B)^\top\|_F$ s.t. $S$ is diagonal: it minimizes the approximation error to $G$ as measured by Frobenius norm (denoted $\|\cdot\|_F$), which KFAC's corresponding $S = S_A \otimes S_B$ cannot generally achieve. A corollary of this is that we will always have $\|G - G_{\text{EKFAC}}\|_F \leq \|G - G_{\text{KFAC}}\|_F$ i.e. EKFAC yields a better approximation of $G$ than KFAC (Theorem 2 proven in Appendix). Figure 2 illustrates the different rescaling strategies, including EKFAC.

**Potential benefits:** Since EKFAC is a better approximation of $G$ than KFAC (in the limited sense of Frobenius norm of the residual) it may yield a better preconditioning of the gradient for optimizing neural networks[3]. Another benefit is linked to computational efficiency: even if KFAC yields a reasonably good approximation in practice, it is costly to re-estimate and invert matrices $A$ and $B$, so this has to be amortized over many updates: re-estimation of the preconditioning is thus typically done at a much lower frequency than the parameter updates, and may lag behind, no longer accurately reflecting the local 2$^{\text{nd}}$ order information. Re-estimating the Kronecker-factored Eigenbasis (KFE)

for EKFAC is similarly costly and must be similarly amortized. But re-estimating the diagonal scaling $s^*$ in that basis is cheap, doable with every mini-batch, so we can hope to reactively track and leverage the changes in $2^{\text{nd}}$ order information along these directions. Figure 3 (right) provides an empirical confirmation that tracking $s^*$ indeed allows to keep the approximation error of $G_{\text{EKFAC}}$ small, compared to $G_{\text{KFAC}}$, between recomputations of basis or inverse .

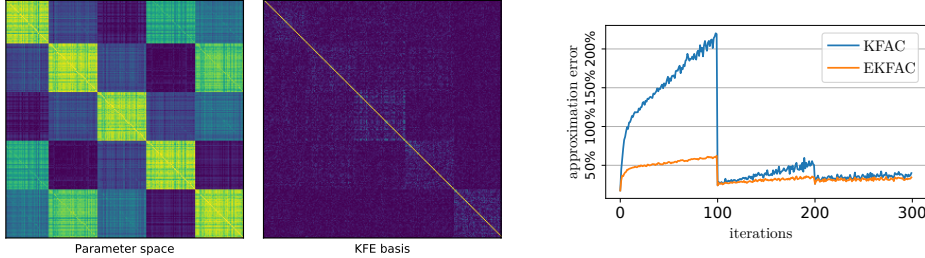

Parameter space           KFE basis

Figure 3: **Left:** Gradient *correlation* matrices measured in the initial parameter basis and in the Kronecker-factored Eigenbasis (KFE), computed from a small 4 sigmoid layer MLP classifier trained on MNIST. Block corresponds to 250 parameters in the 2nd layer. Components are largely decorrelated in the KFE, justifying the use of a diagonal rescaling method *in that basis*.

**Right:** Approximation error $\frac{\|G-\hat{G}\|_F}{\|G\|_F}$ where $\hat{G}$ is either $G_{\text{KFAC}}$ or $G_{\text{EKFAC}}$, for the small MNIST classifier. KFE basis and KFAC inverse are recomputed every 100 iterations. EKFAC's cheap tracking of $s^*$ allows it to drift far less quickly than amortized KFAC from the exact empirical Fisher.

**Dual view by working in the KFE:** Instead of thinking of this new method as an improved factorized approximation of $G$ that we use as a preconditioning matrix, we can alternatively view it as applying a *diagonal method, but in a different basis where the diagonal approximation is more accurate* (an assumption we empirically confirm in Figure 3 left). This can be seen by interpreting the update given by EKFAC as a 3 step process: project the gradient in the KFE (**–**), apply *diagonal* natural gradient descent in this basis (**–**), then project back to the parameter space (**–**):

$$G_{\text{EKFAC}}^{-1} \nabla_\theta = \underbrace{(U_A \otimes U_B)}\, \underbrace{S^{*-1}}\, \underbrace{(U_A \otimes U_B)^\top \nabla_\theta}$$

Note that by writing $\tilde{\nabla}_\theta = (U_A \otimes U_B)^\top \nabla_\theta$ the projected gradient in KFE, the computation of the coefficients $s_i^*$ simplifies in $s_i^* = \mathbb{E}[(\tilde{\nabla}_\theta)_i^2]$. Figure 3 shows gradient correlation matrices in both the initial parameter basis and in the KFE. Gradient components appear far less correlated when expressed in the KFE, which justifies using a diagonal rescaling method *in that basis*.

This viewpoint brings us close to network reparametrization approaches such as Fujimoto & Ohira (2018), whose proposal – that was already hinted towards by Desjardins et al. (2015) – amounts to a reparametrization equivalent of KFAC. More precisely, while Desjardins et al. (2015) empirically explored a reparametrization that uses only input covariance $A$ (and thus corresponds only to "half of" KFAC), Fujimoto & Ohira (2018) extend this to use also backpropagated gradient covariance $B$, making it essentially equivalent to KFAC (with a few extra twists). Our approach differs in that moving to the KFE corresponds to a change of *orthonormal basis*, and more importantly that we *cheaply track* and perform a more optimal *full diagonal* rescaling in that basis, rather than the constrained factored $S_A \otimes S_B$ diagonal that these other approaches are implicitly using.

**Algorithm:** Using Eigenvalue-corrected Kronecker factorization (EKFAC) for neural network optimization involves: a) periodically (every $n$ mini-batches) computing the Kronecker-factored Eigenbasis by doing an eigendecomposition of the same $A$ and $B$ matrices as KFAC; b) estimating scaling vector $s^*$ as second moments of gradient coordinates in that implied basis; c) preconditioning gradients accordingly prior to updating model parameters. Algorithm 1 provides a high level pseudo-code of EKFAC for the case of fully-connected layers[4], and when using it to approximate the empirical Fisher. In this version, we re-estimate $s^*$ from scratch on each mini-batch. An alternative (EKFAC-ra)

is to update $s^*$ as a *running average* of component-wise second moment of mini-batch averaged gradients.

---

**Algorithm 1** EKFAC for fully connected networks

---

**Require:** $n$: recompute eigenbasis every $n$ minibatches
**Require:** $\eta$: learning rate
**Require:** $\epsilon$: damping parameter

**procedure** EKFAC($\mathcal{D}_{\text{train}}$)
    **while** convergence is not reached, iteration $i$ **do**
        sample a minibatch $\mathcal{D}$ from $\mathcal{D}_{\text{train}}$
        Do forward and backprop pass as needed to obtain $h$ and $\delta$
        **for all** layer $l$ **do**
            **if** $i \% n = 0$ **then**          # Amortize eigendecomposition
                COMPUTEEIGENBASIS($\mathcal{D}, l$)
            **end if**
            COMPUTESCALINGS($\mathcal{D}, l$)
            $\nabla^{\text{mini}} \leftarrow \mathbb{E}_{(x,y)\in\mathcal{D}}\left[\nabla_\theta^{(l)}(x,y)\right]$
            UPDATEPARAMETERS($\nabla^{\text{mini}}, l$)
        **end for**
    **end while**
**end procedure**

**procedure** COMPUTEEIGENBASIS($\mathcal{D}, l$)
    $U_A^{(l)}, S_A^{(l)} \leftarrow$ eigendecomposition $\left(\mathbb{E}_\mathcal{D}\left[h^{(l)}h^{(l)\top}\right]\right)$
    $U_B^{(l)}, S_B^{(l)} \leftarrow$ eigendecomposition $\left(\mathbb{E}_\mathcal{D}\left[\delta^{(l)}\delta^{(l)\top}\right]\right)$
**end procedure**

**procedure** COMPUTESCALINGS($\mathcal{D}, l$)
    $s^{*(l)} \leftarrow \mathbb{E}_\mathcal{D}\left[\left(\left(U_A^{(l)} \otimes U_B^{(l)}\right)^\top \nabla_\theta^{(l)}\right)^2\right]$      # Project gradient in eigenbasis[1]
**end procedure**

**procedure** UPDATEPARAMETERS($\nabla^{\text{mini}}, l$)
    $\tilde{\nabla} \leftarrow \left(U_A^{(l)} \otimes U_B^{(l)}\right)^\top \nabla^{\text{mini}}$      # Project gradients in eigenbasis[1]
    $\tilde{\nabla} \leftarrow \tilde{\nabla} / \left(s^{*(l)} + \epsilon\right)$      # Element-wise scaling
    $\nabla^{\text{precond}} \leftarrow \left(U_A^{(l)} \otimes U_B^{(l)}\right)\tilde{\nabla}$      # Project back in parameter basis[1]
    $\theta^{(l)} \leftarrow \theta^{(l)} - \eta\nabla^{\text{precond}}$      # Update parameters
**end procedure**

## 4  Experiments

This section presents an empirical evaluation of our proposed Eigenvalue Corrected KFAC (EKFAC) algorithm in two variants: EKFAC estimates scalings $s^*$ as second moment of intrabatch gradients (in KFE coordinates) as in Algorithm 1, whereas EKFAC-ra estimates $s^*$ as a running average of squared minibatch gradient (in KFE coordinates). We compare them with KFAC and other baselines, primarily SGD with momentum, with and without batch-normalization (BN). For all our experiments KFAC and EKFAC approximate the *empirical Fisher $G$*. This research focuses on improving optimization techniques, so except when specified otherwise, we performed model and hyperparameter selection based on the performance of the optimization objective, i.e. on training loss.

## 4.1 Deep auto-encoder

We consider the task of minimizing the reconstruction error of an 8-layer auto-encoder on the MNIST dataset, a standard task used to benchmark optimization algorithms (Hinton & Salakhutdinov, 2006; Martens & Grosse, 2015; Desjardins et al., 2015). The model consists of an encoder composed of 4 fully-connected sigmoid layers, with a number of hidden units per layer of respectively 1000, 500, 250, 30, and a symmetric decoder (with untied weights).

We compare EKFAC, computing the second moment statistics through its mini-batch, and EKFAC-ra, its running average variant, with different baselines (KFAC, SGD with momentum and BN, ADAM with BN). For each algorithm, best hyperparameters were selected using a mix of grid and random search based on training error. Grid values for hyperparameters are: learning rate $\eta$ and damping $\epsilon$ in $\{10^{-1}, 10^{-2}, 10^{-3}, 10^{-4}\}$, mini-batch size in $\{200, 500\}$. In addition we explored 20 values for $(\eta, \epsilon)$ by random search around each grid points. We found that extra care must be taken when choosing the values of the learning rate and damping parameter $\epsilon$ in order to get good performances, as often observed when working with algorithms that leverage curvature information (see Figure 4 (d)). The learning rate and the damping parameters are kept constant during training.

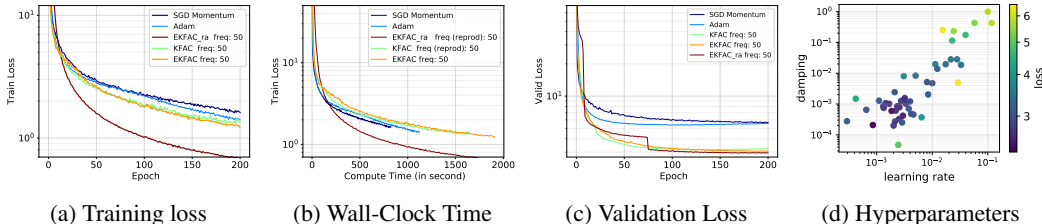

| (a) Training loss | (b) Wall-Clock Time | (c) Validation Loss | (d) Hyperparameters |

Figure 4: MNIST Deep Auto-Encoder task. Models are selected based on the best loss achieved during training. SGD and Adam are with batch-norm. A "freq" of 50 means eigendecomposition or inverse is recomputed every 50 updates. **(a)** Training loss vs epochs. Both EKFAC and EKFAC-ra show an optimization benefit compared to amortized-KFAC and the other baselines. **(b)** Training loss vs wall-clock time. Optimization benefits transfer to faster training for EKFAC-ra. **(c)** Validation performance. KFAC and EKFAC achieve a similar validation performance. **(d)** Sensitivity to hyperparameters values. Color corresponds to final loss reached after 20 epochs for batch size 200.

Figure 4 (a) reports the training loss throughout training and shows that EKFAC and EKFAC-ra both minimize faster the training loss per epoch than KFAC and the other baselines. In addition, Figure 4 (b) shows that the efficient tracking of diagonal scaling vector $s^*$ in EKFAC-ra, despite its slightly increased computational burden per update, allows to achieve faster training in wall-clock time. Finally, on this task, EKFAC and EKFAC-ra achieve better optimization while also maintaining a very good generalization performances (Figure 4 (c)).

Next we investigate how the frequency of the inverse/eigendcomposition recomputation affects optimization. In Figure 5, we compare KFAC/EKFAC with different reparametrization frequencies to a strong KFAC baseline where we reestimate and compute the inverse at each update. This baseline outperforms the amortized version (as a function of number of epochs), and is likely to leverage a better approximation of $G$ as it recomputes the approximated eigenbasis *at each update*. However it comes at a high computational cost, as seen in Figure 5 (b). Amortizing the eigendecomposition allows to strongly decrease the computational cost while slightly degrading the optimization performances. As can be seen in Figure 5 (a), amortized EKFAC preserves better the optimization performance than amortized KFAC. EKFAC re-estimates at each update, the diagonal second moments $s^*$ in the KFE basis, which correspond to the eigenvalues of the EKFAC approximation of $G$. Thus it should better track the true curvature of the loss function. To verify this, we investigate how the eigenspectrum of the true empirical Fisher $G$ changes compared to the eigenspectrum of its approximations as $G_{\text{KFAC}}$ (or $G_{\text{EKFAC}}$). In Figure 5 (c), we track their eigenspectra and report the $l_2$ distance between them during training. We compute the KFE once at the beginning and then keep it fixed during training. We focus on the $4^{\text{th}}$ layer of the auto-encoder: its small size allows to estimate the corresponding $G$ and compute its eigenspectrum at a reasonable cost. We see that the spectrum of $G_{\text{KFAC}}$ quickly diverges from the spectrum of $G$, whereas the cheap frequent reestimation of the diagonal scaling for

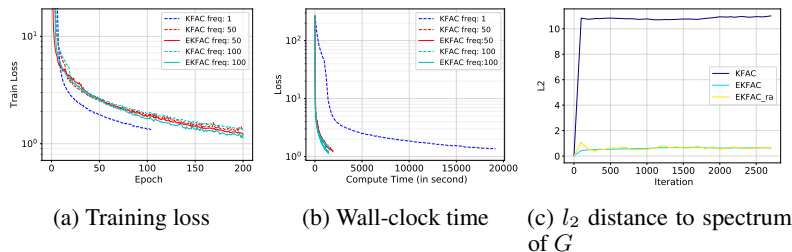

(a) Training loss      (b) Wall-clock time      (c) $l_2$ distance to spectrum of $G$

Figure 5: Impact of frequency of inverse/eigendecomposition recomputation for KFAC/EKFAC. A "freq" of 50 indicates a recomputation every 50 updates. **(a)(b)** Training loss v.s. epochs and wall-clock time. We see that EKFAC preserves better its optimization performances when the eigendecomposition is performed less frequently. **(c)**. Evolution of $l_2$ distance between the eigenspectrum of empirical Fisher $G$ and eigenspectra of approximations $G_{\text{KFAC}}$ and $G_{\text{EKFAC}}$. We see that the spectrum of $G_{\text{KFAC}}$ quickly diverges from the spectrum of $G$, whereas the EKFAC variants, thanks to their frequent diagonal reestimation, manage to much better track $G$.

$G_{\text{EKFAC}}$ and $G_{\text{EKFAC}-\text{ra}}$ allows their spectrum to stay much closer to that of $G$. This is consistent with the evolution of approximation error shown earlier in Figure 3 on the small MNIST classifier.

## 4.2 CIFAR-10

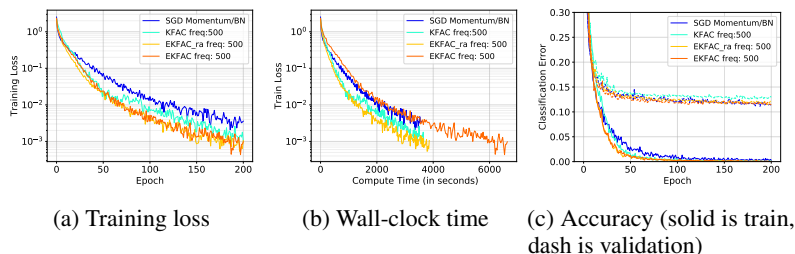

(a) Training loss      (b) Wall-clock time      (c) Accuracy (solid is train, dash is validation)

Figure 6: VGG11 on CIFAR-10. "freq" corresponds to the eigendecomposition (inverse) frequency. In **(a)** and **(b)**, we report the performance of the hyper-parameters reaching the lowest training loss for each epoch (to highlight which optimizers perform best given a fixed epoch budget). In **(c)** models are selected according to the best overall validation error. When the inverse/eigendecomposition is amortized on 500 iterations, EKFAC-ra shows an optimization benefit while maintaining its generalization capability.

In this section, we evaluate our proposed algorithm on the CIFAR-10 dataset using a VGG11 convolutional neural network (Simonyan & Zisserman, 2015) and a Resnet34 (He et al., 2016). To implement KFAC/EKFAC in a convolutional neural network, we rely on the SUA approximation (Grosse & Martens, 2016) which has been shown to be competitive in practice (Laurent et al., 2018). We highlight that we do not use BN in our model when they are trained using KFAC/EKFAC.

As in the previous experiments, a grid search is performed to select the hyperparameters. Around each grid point, learning rate and damping values are further explored through random search. We experiment with constant learning rate in this section, but explore learning rate schedule with KFAC/EKFAC in Appendix D.2. the damping parameter is initialized according to Appendix C. In the figures that show the model training loss per epoch or wall-clock time, we report the performance of the hyper-parameters attaining the lowest training loss for each epoch. This per-epoch model selection allows to show which model type reaches the lowest cost during training and also which one optimizes best given any "epoch budget". We did not find one single set of hyperparameter for which the EKFAC optimization curve is below KFAC for all the epochs (and vice-versa). However, doing a per-epoch model selection shows that the best EKFAC configuration usually outperforms the best KFAC for any chosen target epoch. This is also true for any chosen compute time budget.

In Figure 6, we compare EKFAC/EKFAC-ra to KFAC and SGD Momentum with or without BN when training a VGG-11 network. We use a batch size of 500 for the KFAC based approaches and 200 for the SGD baselines. Figure 6 (a) show that EKFAC yields better optimization than the SGD baselines and KFAC in training loss per epoch when the computation of the KFE is amortized. Figure 6 (c) also shows that models trained with EKFAC maintain good generalization. EKFAC-ra shows some wall-clock time improvements over the baselines in that setting ( Figure 6 (b)). However, we observe that using KFAC with a batch size of 200 can catch-up with EKFAC (but not EKFAC-ra) in wall-clock time despite being outperformed in term of optimization per iteration (see Figure D.2, in Appendix). VGG11 is a relatively small network by modern standard and the KFAC (with SUA approximation) remains computationally bearable in this model. We hypothesize that using smaller batches, KFAC can be updated often enough per epoch to have a reasonable estimation error while not paying too high a computational price.

In Figure 7, we report similar results when training a Resnet34. We compare EKFAC-ra with KFAC, and SGD with momentum and BN. To be able to train the Resnet34 without BN, we need to rely on a careful initialization scheme (detailed in Appendix B) in order to ensure good signal propagation during the forward and backward passes. EKFAC-ra outperforms both KFAC (when amortized) and SGD with momentum and BN in term of optimization per epochs, and compute time. This gain appears robust across different batch sizes (see Figure D.3 in the Appendix).

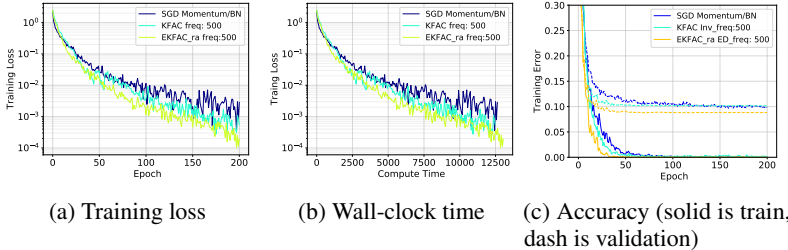

(a) Training loss     (b) Wall-clock time     (c) Accuracy (solid is train, dash is validation)

Figure 7: Training a Resnet Network with 34 layers on CIFAR-10. "freq" corresponds to eigende-composition (inverse) frequency. In **(a)** and **(b)**, we report the performance of the hyper-parameters reaching the lowest training loss for each epoch (to highlight which optimizers perform best given a fixed epoch budget). In **(c)** we select model according to the best overall validation error. When the inverse/eigen decomposition is amortized on 500 iterations, EKFAC-ra shows optimization and computational time benefits while maintaining a good generalization capability.

## 5   Conclusion and future work

In this work, we introduced the Eigenvalue-corrected Kronecker factorization (EKFAC), an approximate factorization of the (empirical) Fisher Information Matrix that is computationally manageable while still being accurate. We formally proved (in Appendix) that EKFAC yields a more accurate approximation than its closest parent and competitor KFAC, in the sense of the Frobenius norm. Of more practical importance, we showed that our algorithm allows to cheaply perform partial updates of the curvature estimate, by maintaining an up-to-date estimate of its eigenvalues while keeping the estimate of its eigenbasis fixed. This partial updating proves competitive when applied to optimizing deep networks, both with respect to the number of iterations and wall-clock time.

Our approach amounts to normalizing the gradient by its $2^{nd}$ moment component-wise in a Kronecker-factored Eigenbasis (KFE). But one could apply other component-wise (diagonal) adaptive algorithms such as Adagrad (Duchi et al., 2011), RMSProp (Tieleman & Hinton, 2012) or Adam (Kingma & Ba, 2015), *in the KFE* where the diagonal approximation is much more accurate. This is left for future work. We also intend to explore alternative strategies for obtaining the approximate eigenbasis and investigate how to increase the robustness of the algorithm with respect to the damping hyperparameter. We also want to explore novel regularization strategies, so that the advantage of efficient optimization algorithms can more reliably be translated to generalization error.

**Acknowledgments**

The experiments were conducted using PyTorch (Paszke et al. (2017)). The authors would like to thank Facebook, CIFAR, Calcul Quebec and Compute Canada, for research funding and computational resources.

## Footnotes

[1]Since $(A \otimes B)^{-1} = A^{-1} \otimes B^{-1}$.

[2]This approximation is done separately for each block $G^{(l)}$, we dropped the superscript to simplify notations.

[3]Although there is no guarantee. In particular $G_{\text{EKFAC}}$ being a better approximation of $G$ does not guarantee that $G_{\text{EKFAC}}^{-1}\nabla_\theta$ will be closer to the natural gradient update direction $G^{-1}\nabla_\theta$ .

[4]EKFAC for convolutional layers follows the same structure, but require a more convoluted notation.

[1] Can be efficiently computed using the following identity: $(A \otimes B)\text{vec}(C) = B^\top C A$

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
