[Supplementary Material · paper_with_appendix.pdf]

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

# A  Proofs

## A.1  Proof that EKFAC does optimal diagonal rescaling in the KFE

**Lemma 1.** *Let $G = \mathbb{E}\left[\nabla_\theta \nabla_\theta^\top\right]$ a real positive semi-definite matrix. And let $Q$ a given orthogonal matrix. Among diagonal matrices, diagonal matrix $D$ with diagonal entries $D_{ii} = \mathbb{E}\left[\left(Q^\top \nabla_\theta\right)_i^2\right]$ minimize approximation error $e = \left\|G - QDQ^\top\right\|_F$ (measured as Frobenius norm).*

*Proof.* Since the Frobenius norm remains unchanged through multiplication by an orthogonal matrix we can write

$$
\begin{aligned}
e^2 &= \left\|G - QDQ^\top\right\|_F^2 \\
&= \left\|Q^\top \left(G - QDQ^\top\right) Q\right\|_F^2 \\
&= \left\|Q^\top GQ - D\right\|_F^2 \\
&= \underbrace{\sum_i (Q^\top GQ - D)_{ii}^2}_{\text{diagonal}} + \underbrace{\sum_i \sum_{j \neq i} (Q^\top GQ)_{ij}^2}_{\text{off-diagonal}}
\end{aligned}
$$

Since $D$ is diagonal, it does not affect the off-diagonal terms.

The squared diagonal terms all reach their minimum value 0 by setting $D_{ii} = \left(Q^\top GQ\right)_{ii}$ for all $i$:

$$
\begin{aligned}
D_{ii} &= \left(Q^\top GQ\right)_{ii} \\
&= \left(Q^\top \mathbb{E}\left[\nabla_\theta \nabla_\theta^\top\right] Q\right)_{ii} \\
&= \left(\mathbb{E}\left[Q^\top \nabla_\theta \nabla_\theta^\top Q\right]\right)_{ii} \\
&= \left(\mathbb{E}\left[Q^\top \nabla_\theta \left(Q^\top \nabla_\theta\right)^\top\right]\right)_{ii} \\
&= \mathbb{E}\left[\left(Q^\top \nabla_\theta\right)_i^2\right] \text{ since } Q^\top \nabla_\theta \text{ is a vector}
\end{aligned}
$$

We have thus shown that diagonal matrix $D$ with diagonal entries $D_{ii} = \mathbb{E}\left[\left(Q^\top \nabla_\theta\right)_i^2\right]$ minimize $e^2$. Since Frobenius norm $e$ is non-negative this implies that $D$ also minimizes $e$. □

**Theorem 2.** *Let $G = \mathbb{E}\left[\nabla_\theta \nabla_\theta^\top\right]$ the matrix we want to approximate. Let $G_{\mathrm{KFAC}} = A \otimes B$ the approximation of $G$ obtained by KFAC. Let $A = U_A S_A U_A^\top$ and $B = U_B S_B U_B^\top$ eigendecomposition of $A$ and $B$. The diagonal rescaling that EKFAC does in the Kronecker-factored Eigenbasis (KFE) $U_A \otimes U_B$ is optimal in the sense that it minimizes the Frobenius norm of the approximation error: among diagonal matrices $D$, approximation error $e = \left\|G - (U_A \otimes U_B)D(U_A \otimes U_B)^\top\right\|_F$ is minimized by the matrix with with diagonal entries $D_{ii} = s_i^* = \mathbb{E}\left[\left((U_A \otimes U_B)^\top \nabla_\theta\right)_i^2\right]$.*

*Proof.* This follows directly by setting $Q = U_A \otimes U_B$ in Lemma 1. Note that the Kronecker product of two orthogonal matrices yields an orthogonal matrix.

□

**Theorem 3.** *Let $G_{\mathrm{KFAC}}$ the KFAC approximation of $G$ and $G_{\mathrm{EKFAC}}$ the EKFAC approximation of $G$, we always have $\|G - G_{\mathrm{EKFAC}}\|_F \leq \|G - G_{\mathrm{KFAC}}\|_F$.*

*Proof.* This follows trivially from Theorem 2 on the optimality of the EKFAC diagonal rescaling.

Since $D = S^*$, with the $S^* = \mathrm{diag}(s^*)$ of EKFAC, minimizes $\left\|G - (U_A \otimes U_B)D(U_A \otimes U_B)^\top\right\|_F$, it implies that:

$$
\begin{aligned}
\|G - (U_A \otimes U_B)S^*(U_A \otimes U_B)^\top\|_F &\leq \|G - (U_A \otimes U_B)(S_A \otimes S_B)(U_A \otimes U_B)^\top\|_F \\
\|G - G_{\mathrm{EKFAC}}\|_F &\leq \|G - (U_A S_A U_A^\top) \otimes (U_B S_B U_B^\top)\|_F \\
\|G - G_{\mathrm{EKFAC}}\|_F &\leq \|G - A \otimes B)\|_F \\
\|G - G_{\mathrm{EKFAC}}\|_F &\leq \|G - G_{\mathrm{KFAC}}\|_F
\end{aligned}
$$

$\square$

We have thus demonstrated that EKFAC yields a better approximation (more precisely: at least as good in Frobenius norm error) of $G$ than KFAC.

## A.2 Proof that $S_{ii} = \mathbb{E}\left[\left(U^\top \nabla_\theta\right)_i^2\right]$

**Theorem 4.** *Let $G = \mathbb{E}\left[\nabla_\theta \nabla_\theta^\top\right]$ and $G = USU^\top$ its eigendecomposition. Then $S_{ii} = \mathbb{E}\left[\left(U^\top \nabla_\theta\right)_i^2\right]$.*

*Proof.* Starting from eigendecomposition $G = USU^\top$ and the fact that $U$ is orthogonal so that $U^\top U = I$ we can write

$$
\begin{aligned}
G &= USU^\top \\
U^\top G U &= U^\top USU^\top U \\
U^\top \underbrace{G}_{\mathbb{E}[\nabla_\theta \nabla_\theta^\top]} U &= S
\end{aligned}
$$

so that

$$
\begin{aligned}
S_{ii} &= \left(U^\top \mathbb{E}\left[\nabla_\theta \nabla_\theta^\top\right] U\right)_{ii} \\
&= \left(\mathbb{E}\left[U^\top \nabla_\theta \nabla_\theta^\top U\right]\right)_{ii} \\
&= \left(\mathbb{E}\left[U^\top \nabla_\theta \left(U^\top \nabla_\theta\right)\right]\right)_{ii} \\
&= \mathbb{E}\left[\left(U^\top \nabla_\theta \left(U^\top \nabla_\theta\right)\right)_{ii}\right] \\
&= \mathbb{E}\left[\left(U^\top \nabla_\theta\right)_i^2\right]
\end{aligned}
$$

where we obtained the last equality by observing that $U^\top \nabla_\theta$ is a vector and that the diagonal entries of the matrix $aa^\top$ for any vector $a$ are given by $a^2$ where the square operation is element-wise. $\square$

## B Residual network initialization

To train residual networks without using BN, one need to initialize them carefully, so we used the following procedure, denoting $n$ the fan-in of the layer:

1. We use the He initialization for each layer directly preceded by a ReLU (He et al., 2015): $W \sim \mathcal{N}(0, 2/n)$, $b = 0$.

2. Each layer not directly preceded by an activation function (for example the convolution in a skip connection) is initialized as: $W \sim \mathcal{N}(0, 1/n)$, $b = 0$. This can be derived from the He initialization, using the Identity as activation function.

3. Inspired from Goyal et al. (2017), we divide the scale of the last convolution in each residual block by a factor 10: $W \sim \mathcal{N}(0, 0.2/n)$, $b = 0$. This not only helps preserving the variance through the network but also eases the optimization at the beginning of the training.

## C Initialization of $\epsilon$

Both KFAC and EKFAC are very sensitive to the the damping parameter $\epsilon$. When EKFAC is applied to deep neural networks, we empirically observe that adding $\epsilon$ to the eigenvalues of both the activation covariance and gradient covariance matrices lead to better optimization than using a fixed $\epsilon$ added directly on the eigenvalues in KFE space. KFAC uses also a similar initialization for $\epsilon$. By setting $\text{diag}(\epsilon') = \epsilon I + S_A \otimes \sqrt{\epsilon} I_{d_{\text{out}}} + \sqrt{\epsilon} I_{d_{\text{in}}} \otimes S_B$ , we can implement this initialization strategy directly in the KFE.

# D   Additional empirical results

## D.1   Impact of batch size

In this section, we evaluate the impact of the batch size on the optimization performances for KFAC and EKFAC. In Figure D.1, we reports the training loss performance per epochs for different batch sizes for VGG11. We observe that the optimization gain of EKFAC with respect of KFAC diminishes as the batch size gets smaller.

(a) Batch size = 200          (b) Batch size = 500          (c) Batch size = 1000

Figure D.1: VGG11 on CIFAR-10. ED_freq (Inv_freq) corresponds to eigendecomposition (inverse) frequency. We perform model selections according to best training loss at each epoch. On this setting, we observe that the optimization gain of EKFAC with respect of KFAC diminishes as the batch size reduces.

In Figure D.2, we look at the training loss per iterations and the training loss per computation time for different batch sizes, again on VGG11. EKFAC shows optimization benefits over KFAC as we increase the batch size (thus reducing the number of inverse/eigendecomposition per epoch). This gain does not translate in faster training in term of computation time in that setting. VGG11 is a relatively small network by modern standard and the SUA approximation remains computationally bearable on this model. We hypothesize that using smaller batches, KFAC can be updated often enough per epoch to have a reasonable estimation error while not paying a computational price too high.

(a) Iterations          (b) Wall-clock time

Figure D.2: VGG11 on CIFAR-10. ED_freq (Inv_freq) corresponds to eigendecomposition (inverse) frequency. We perform model selections according to best training loss at each epoch. **(a)** Training loss per iterations for different batch sizes. **(b)** Training loss per computation time for different batch sizes. EKFAC shows optimization benefits over KFAC as we increases the batch size (thus reducing the number of inverse/eigendecomposition per epoch). This gain does not translate in faster training in terms of wall-clock time in that setting.

In Figure D.3, we perform a similar experiment on the Resnet34. In this setting, we observe that the optimization gain of EKFAC with respect of KFAC remains consistent across batch sizes.

## D.2   Learning rate schedule

In this section we investigate the impact of a learning rate schedule on the optimization of EKFAC. We use a similar setting than the CIFAR-10 experiment, except that we decay the learning by 2 every 20 epochs. Figure D.4 and D.5 show that EKFAC still highlight some optimization benefit, relatively to the baseline, when combined with a learning rate schedule.

(a) Batch size = 200        (b) Batch size = 500

Figure D.3: Resnet34 on CIFAR-10. ED_freq (Inv_freq) corresponds to eigendecomposition (inverse) frequency. We perform model selections according to best training loss at each epoch. In this setting, we observe that the optimization gain of EKFAC with respect of KFAC remains consistent across batch sizes.

(a) Training loss        (b) Wall-clock time

Figure D.4: VGG11 on CIFAR10 using a learning rate schedule. ED_freq (Inv_freq) corresponds to eigendecomposition (inverse) frequency. In (a) and (b), we report the performance of the hyper-parameters reaching the lowest training loss for each epoch (to highlight which optimizers perform best given a fixed epoch budget).

(a) Training loss        (b) Wall-clock time

Figure D.5: Resnet34 on CIFAR10 using a learning rate schedule. ED_freq (Inv_freq) corresponds to eigendecomposition (inverse) frequency. In (a) and (b), we report the performance of the hyper-parameters reaching the lowest training loss for each epoch (to highlight which optimizers perform best given a fixed epoch budget).