[Reviews · NeurIPS 2018]

Reviewer 1



The authors make an interesting observation that KFAC approximation of the Fisher information matrix results in a pre-conditioner with potentially mis-scaled variances of the gradient in the Kronecker-factorized basis. The paper propose to fix the issue by correcting the eigenspectrum of the KFAC pre-conditioner and develops a simple extension of the KFAC algorithm, termed EKFAC. The paper is well written, easy to follow, and makes an interesting observation. A few comments / questions for the authors: - As noted in line 133, the original KFAC rescaling is not guaranteed to match the second moments of the correct preconditioner, it would be interesting to see when that happens and much the off. Is there a way to bound error due to the approximation? - Related to previous point, would be nice to see quantitative differences between the spectra of the preconditioners computed by KFAC vs. EKFAC on a toy problem where one can compute correct preconditioners exactly/analytically. - The paper applied EKFAC to MLPs and convnets. Should one expect any improvements when using the method for training recurrent networks?

Reviewer 2



Summary The paper describes a generic 2nd order stochastic optimisation scheme exploiting curvature information to improve the trade-off between convergence speed und computational effort. It proposes an extension to the approximate natural gradient method KFAC where the Fisher information matrix is restricted to be of Kronecker structure. The authors propose to relax the Kronecker constraint and suggest to use a general diagonal scaling matrix rather than a diagonal Kronecker scaling matrix. This diagonal scaling matrix is estimated from gradients along with the Kronecker eigenbasis. Quality The idea in the paper is convincing and makes sense. The derivation in the appendix looks correct. Although the empirical evaluation could have included some more baseline methods, the paper is in a very good state and contains all the information required to turn the idea into an implementation and to judge its merits. Clarity The paper is well written, the figures are crisp and since the idea is technically simple; the material is easy to grasp. Originality It seems that the proposed extension was not proposed before but it seems like a straight forward extension. Significance The extension improves upon the basic KFAC algorithm and hence extends the optimisation toolbox of neural network practicioners. Empirical Evaluation There are two sets of experiments, an MNIST autoencoder and VGG11 on CIFAR-10. The plots illustrate that there are situations where the extensions yields a favorable outcome compared to plain KFAC and SGD. Unfortunately, there is no curve for a simple diagonal scaling algorithm such as batch normalization or RMSprop, which renders the evaluation slightly incomplete. Reproducibility The authors do not provide code but a detailed algorithm box in pseudo code. The datasets MNIST and CIFAR-10 are public and the network architectures are standard. So it should be possible to reproduce the results. Details and Typos - line 63: not sure why there is an $f_theta$ - line 73: "Thus, natural" I've read the rebuttal and updated the score and evaluation accordingly based on the discussion.

Reviewer 3



This paper proposes a new Kronecker approximation of the empirical Fisher information, EKFAC, that matches the eigenvalues of the Fisher in the approximation; this method is then used for natural gradient descent like the original Kronecker factored approximate curvature method (KFAC). The authors claim that the proposed method has both enhanced computational benefits over the original KFAC scheme and increased accuracy than the previous KFAC variants. However, it does not seem to outperform well-tuned SGD with momentum and batch norm and has approximately double the computational costs according to the figures in the paper. The tradeoff with the proposed method seems to be a marginal increase in test accuracy at the expense of a significant increase in computational effort. Overall, the experiments section is quite confusing to me and the figures do not seem to align well with the descriptions in the text of the paper. Originality: The proposed method seems to be quite novel. Instead of directly computing an approximation to the empirical Fisher information, EKFAC computes an approximation to the eigen-decomposition. This has the interpretation of projecting the gradients into an eigen-basis, performing natural gradient descent in the eigen-space, before projecting back into the standard weight space. Significance: The task of developing new second-order stochastic optimization methods that scale to deep neural networks is quite significant. The proposed method seems to be one of the first methods to use an eigen-decomposition to perform natural gradient descent. However, the improvement in practice seems to be quite minimal when compared to SGD and SGD + momentum. Clarity: The paper is relatively well-written, with the exception of the experiments section (see below). However, I would suggest more paragraph breaks, particularly in Section 1 and Section 3.1, to further encourage readability. Quality: The method developed seems to be sound. However, I have some issues with the experiment section (see below), particularly with respect to the presentation. Much emphasis is placed on the wall-clock time, but no description of the system architectures or units on the time (seconds? minutes?) is mentioned. Comments on Experiments: - The color scheme in the plots with respect to the SGD/Momentum/Adam variants is quite unclear. For instance, in Figure 4c, it is obvious that one of these baseline optimizers gives quite similar validation loss as KFAC and EKFAC but noticing this is hindered by both the color scheme and the KFAC loss curve lines being on top of that curve. This is true for Figure 6 as well, particularly Figure 6c, which has the additional complexity of having dashed lines for the test set in the same plot as the train set accuracy. Perhaps tables could be used to report accuracy and wall-clock time instead. - Figure 4b: Could the authors comment on why the two EKFAC_ra curves (yellow and orange) are by far the slowest methods on the deep auto-encoding task? By contrast, in the text, EKFAC_ra is described as a quicker than the EKFAC and KFAC methods. - Figure 5a and 5b: From these figures, it is relatively clear that KFAC performs better with respect to training loss when updated more frequently; the trade-off is of course computation time. However, there is no line for EKFAC updated every 1 iteration (like for KFAC). This makes it hard to substantiate the claim that “EKFAC preserves better its optimization performances as we decrease the frequency of eigendecomposition,” as there are only two data points for EKFAC (50 and 100). Additionally, could the authors comment on if the running average version of EKFAC should also maintain this performance as it is not in this plot? - Figure 6b and 6d: Could the authors please comment on the vastly improved speed of KFAC (yellow line) in terms of wall-clock time on the CIFAR 10 experiment? From these figures, it seems like KFAC requires a similar amount of run time to SGD + momentum, and considerably less wall-clock time than EKFAC. - A suggestion for another experiment: what is the accuracy on a problem when comparing SGD (and friends) to EKFAC for the same amount of wall-clock time? - Finally, could the authors test the accuracy of the KFAC/EKFAC approximations on a small MLP (say one or two hidden layers and 20 hidden units) where the empirical Fisher information and its eigenvalues can be computed exactly? This might be more effective in showing that EKFAC is more accurate than KFAC. The experiment with the fourth layer of the deep autoencoder is quite nice; however, it would be interesting to see how EKFAC and KFAC model the cross-correlations between the two sets of weight parameters. Updates after Author Response: I thank the authors for their clarifications, particularly with respect my comments on the figures in the main paper. I do hope that all of their updated figures look as interpretable as the ones that they showed in the response. As a small comment, I would also like to see the error figure that they included in the response in the main paper (and with more y-axis labels), as it shows the error associated with using KFAC and EKFAC very nicely. However, I'm still unconvinced by the author's claims that EKFAC_ra is slower in convnets due to issues with computing a running average of the second moment of gradients, as this seems like it would just require a square operation. Typos: Line 25: 1st -> 1st order Line 28: mention that \eta is the learning rate. Line 34: Diagonal approximate -> Diagonal approximations Line 35: are proven to be efficient -> have been proven to be efficient Line 38: view) -> view Line 54: while computing Kronecker-factored eigenbasis is an expensive operation that need to be -> while computing [a] Kronecker-factored eigenbasis is an expensive operation that [needs] to be Line 75: Whereas … uses is not a sentence and should probably be joined with the previous sentence. Alternatively, change Whereas to (By contrast,). Lines 61-80: This paragraph seems to be quite run-on and should be separated into two separate paragraphs. Line 230: missing . after (Laurent et al, 2018).